# Biomechanical, Anthropometric and Psychological Determinants of Barbell Bench Press Strength

**DOI:** 10.3390/sports10120199

**Published:** 2022-12-05

**Authors:** Derrick W. Van Every, Max Coleman, Daniel L. Plotkin, Hugo Zambrano, Bas Van Hooren, Stian Larsen, Greg Nuckols, Andrew D. Vigotsky, Brad J. Schoenfeld

**Affiliations:** 1Department of Health Sciences, CUNY Lehman College, New York, NY 10468, USA; 2Department of Nutrition and Movement Sciences, NUTRIM School of Nutrition and Translational Research in Metabolism, Maastricht University Medical Centre+, 6229 HX Maastricht, The Netherlands; 3Department of Sports Sciences and Physical Education, Nord University, 7600 Levanger, Norway; 4Stronger by Science LLC, Raleigh, NC 27609, USA; 5Departments of Biomedical Engineering and Statistics, Northwestern University, Evanston, IL 60208, USA

**Keywords:** kinematics, kinetics, one repetition maximum, pectoralis major

## Abstract

The purpose of this study was to improve our understanding of the relative contributions of biomechanical, anthropometric, and psychological factors in explaining maximal bench press (BP) strength in a heterogeneous, resistance-trained sample. Eighteen college-aged participants reported to the laboratory for three visits. The first visit consisted of psychometric testing. The second visit assessed participants’ anthropometrics, additional psychometric outcomes, and bench press one repetition maximum (1RM). Participants performed isometric dynamometry testing for horizontal shoulder adduction and elbow extension at a predicted sticking point joint position. Multiple linear regression was used to examine the relationships between the biomechanical, anthropometric, and psychological variables and BP 1RM. Our primary multiple linear regression accounted for 43% of the variance in BP strength (F(3,14) = 5.34, *p* = 0.01; R^2^ = 0.53; adjusted R^2^ = 0.43). The sum of peak isometric net joint moments from the shoulder and elbow had the greatest standardized effect (0.59), followed by lean body mass (0.27) and self-efficacy (0.17). The variance in BP 1RM can be similarly captured (R^2^ = 0.48) by a single principal component containing anthropometric, biomechanics, and psychological variables. Pearson correlations with BP strength were generally greater among anthropometric and biomechanical variables as compared to psychological variables. These data suggest that BP strength among a heterogeneous, resistance-trained population is explained by multiple factors and is more strongly associated with physical than psychological variables.

## 1. Introduction

The barbell bench press (BP) is widely used to improve the strength and power of the upper body [1,2,3] and is one of three exercises performed in the sport of powerlifting. As such, many lifters—ranging from recreational to competitive powerlifters—aim to improve their BP strength [4]. Therefore, a better understanding of the determinants of BP strength may have important practical implications. In principle, by understanding the determinants of BP strength, practitioners can improve training regimens aiming to maximize BP strength. In the lab, a better understanding of BP performance may facilitate the development and evaluation of novel resistance training interventions to improve the BP.

Many factors determine BP performance, yet their relative importance remains to be explored. Since the force-producing ability of a muscle is proportional to the number of sarcomeres in parallel [5], a greater magnitude of muscle mass should confer a strength advantage. In line with this idea, fat-free mass (FFM) and muscle mass are strongly, positively correlated with BP performance [6,7,8], suggesting that maximum BP strength may be limited by an individual’s skeletal muscle mass [6]. Although the relationship between muscle mass and strength may seem straightforward, individual differences in muscle geometry and architecture may complicate matters [5,9].

To help resolve some of the variation in how muscle mass may confer an advantage, one can assess a different but related construct, joint strength (see [10]), which would dictate the biomechanical constraints within which each joint must work to complete the BP. However, due to the dynamic and multi-joint nature of the BP, neither FFM nor joint strength is likely sufficient to fully explain individual differences in BP performance. In the squat, which is analogous to the BP in terms of being a compound movement, Vigotsky et al. observed that net joint moments did not approach 100% of what each joint was capable of in isolation, suggesting that individuals may be less than the sum of their parts when it comes to multi-joint force production [10]. Conceivably, given that the BP requires simultaneous shoulder flexion and elbow extension, humans are likely to produce greater net joint moments in isolation than during the dynamic exercise. Notwithstanding, single-joint strength should serve as a biomechanical constraint within which complex, multi-joint movements can be performed. These biomechanical constraints may serve as a basis for understanding the combined actions of a given compound movement.

Although a handful of studies have investigated neuromechanical and anthropometric determinants of BP strength [7,11,12,13], psychological determinants have largely been ignored. One such psychological variable of interest is self-efficacy—one’s belief in oneself to execute a task or attain certain performance outcomes [14]. Since self-efficacy plays a role in strength and physical performance [15,16,17], it may indeed represent another dimension or determinant of BP performance. Thus, it is prudent to investigate both physical and psychological variables to fully understand BP strength.

The purpose of this paper was to investigate biomechanical, anthropometric, and psychological determinants of maximum barbell BP strength in a heterogeneous sample of young, resistance-trained men and women. We hypothesized that the anthropometric, biomechanical, and psychological variables would all be independently associated with BP performance, and we predicted that the combination of anthropometric, biomechanical, and psychological variables would explain a majority of the variance in BP performance.

## 2. Methods

### 2.1. Experimental Approach to the Problem

In a cross-sectional design, participants reported to the human performance laboratory for 3 visits, with the second and third visit separated by a minimum of 48 h. During the first visit, participants completed psychological assessments including the Physical Self-Efficacy survey and a BP self-efficacy survey. The second visit included anthropometric measurements, followed by a second BP self-efficacy survey and BP 1RM testing. Three-dimensional (3D) motion capture (Vicon Nexus, v2.14, Vicon, Oxford, UK) was used to analyze the barbell trajectory, from which sticking point joint angles were derived for dynamometry testing. In addition, force plates were used to measure ground reaction forces at the feet. During the third visit, participants’ isometric joint strength was tested on a dynamometer, using each participant’s predicted joint angles. The data were analyzed using linear regression to elucidate the determinants of barbell BP strength.

### 2.2. Participants

To identify the determinants of barbell BP strength, a convenience sample of males and females (aged 18–35 years) was recruited from a university population. A total of 18 male and female volunteers (male *n* = 16; female *n* = 2) participated in the study. Participant demographics can be viewed in Table 1. As determined by Monte Carlo analysis, this sample provides sufficient precision (±CI_95%_) to rule out smaller correlations (≤0.3) when the effect is large (e.g., *r* ≥ 0.7) and to rule out larger correlations (≥0.5) when the effect is null. An effect size equivalent to a Pearson’s *r* of 0.7 was conservatively chosen, as previous research revealed correlation coefficients greater than 0.8 for one of the primary outcomes (relative fat-free mass) with BP performance [8]. Inclusion criteria were (1) a minimum of 1 year of resistance training experience, including regular performance of the BP (at least once per week for the past 6 months); (2) free from injury or illness potentially impacting their participation, or that may potentially be worsened by their performance in the BP; (3) self-reportedly free from the use of anabolic steroids within the past year. To account for the use of bioelectrical impedance testing, participants were excluded if they were pregnant, had any limb amputations, or any electronic implants (i.e., heart pacemaker and brain stimulator). All participants were required to answer “no” to all questions on the Physical Activity Readiness Questionnaire. Participants were instructed to avoid any upper-body resistance training for 48 h before their second and third laboratory visits. Approval for the study was obtained from the Institutional Review Board of the City University of New York, Lehman College. Participants provided written informed consent to acknowledge they were apprised of the potential risks and benefits of participation. The methods for this study were preregistered on the Open Science Framework prior to recruitment (accessed on 22 February 2022).

### 2.3. Procedures

#### 2.3.1. Anthropometric and Body Composition

Measures of body mass and height were obtained using a digital scale (InBody 770, Biospace Co. Ltd., Seoul, Republic of Korea) and stadiometer (DETECTO USA, Webb City, MO, USA), respectively. Moreover, measures of the lengths of the upper arm and forearm, width of the shoulders and hips, chest depth, arm span, and circumferences (forearm, relaxed and flexed upper arm, chest, stomach, hips, thighs, and shanks) were recorded using a flexible tape measure. Each measurement was taken bilaterally by the same investigator and recorded twice, then averaged to obtain a final value; in the case of a discrepancy greater than 5%, a third measurement was taken, and all 3 measurements were averaged. In an effort to compare our results to the findings of Reya et al. [12], we calculated the following anthropometric ratios for each participant as reported by Reya et al. [12]: (1) Brugsch index—chest circumference divided by height; (2) ilio-acromial index—iliac width divided by acromial width; (3) brachial index—the length of the forearm divided by the length of the humerus; (4) arm length to body height ratio.

Measurements of fat mass, fat-free mass (FFM), segmental body mass, and body water content were collected using an InBody 770 multifrequency bioelectrical impedance (BIA) device (Biospace Co. Ltd., Seoul, Republic of Korea) according to the manufacturer’s instructions. Participants were asked to refrain from eating 12 h prior to testing, and to eliminate alcohol consumption and avoid strenuous exercise for 24 h. Additionally, participants were asked to avoid drinking fluids the morning of testing and to void their bladder immediately prior to the test. After cleansing their hands and feet with a sanitary wipe, participants were required to stand on the unit with their heels centered on the electrodes, grasp the handles of the unit, abduct their arms approximately 30 degrees and remain as motionless as possible while the unit estimated body composition. Previous investigations have reported good agreement from BIA to a four-compartment model [18] and it has been deemed a reliable proxy for the estimation of body composition in normal-weight adults in the absence of dual X-ray absorptiometry [19].

#### 2.3.2. Kinetic/Kinematic Analysis

Kinematic measures of the barbell were obtained using 6 Vicon 3D Vero cameras (Vicon Nexus, v2.14, Vicon, Oxford, UK) that track the *x*-, *y*-, and *z*-coordinates of reflective markers at a frequency of 100 Hz. Reflective markers were placed in the center and on both ends of the barbell, the positions of which were exported to a comma separated values file (.csv) for analysis. The kinematics of the bar were used to identify distinct phases during the BP, which were denoted as follows: (a) onset of ascent (concentric), the minimum vertical position of the barbell before an upward change in direction; (b) sticking point (minimum vertical bar acceleration); and (c) the end of ascent (lockout), maximum vertical position prior to re-racking the barbell.

As the ability to overcome the sticking region is likely the primary technical factor in determining BP 1RM [20], the kinematic factors were analyzed during the point of maximum bar deceleration during the ascent. Estimates of anatomical joint angles were then calculated by a predictive model (R, R Core Team, Vienna, Austria; version 4.2.1). Specifically, using 9 individuals’ data, we built an L_1_-regularized multivariate regression to predict sticking point elbow and shoulder angles from upper arm length and percentage of the concentric phase when minimum bar acceleration occurred. Hyperparameter λ was chosen using leave-one-out cross-validation, and expected root mean squared errors (RMSEs) were estimated using the 0.632+ bootstrap (6.6°, 7.3°, and 13.2° for shoulder flexion, shoulder abduction, and elbow flexion, respectively). Lastly, peak bilateral ground reaction forces of the feet during the 1RM BP were measured using dual force plates (Model 9260AA, Kistler Group, Winterthur, Switzerland; sampling at 100 Hz) to help make inferences about contributions from “leg drive”.

#### 2.3.3. Physical Self-Efficacy Survey

To measure self-efficacy, participants were asked to complete a physical self-efficacy (PSE) survey, which has been shown to have a test–retest intraclass correlation coefficient (ICC) of 0.80 when administered 6 weeks apart [21]. The PSE consists of 22 questions, each on a 6-point Likert scale with 2 subscales: perceived physical ability (score range = 10–60) and physical self-presentation confidence (score range = 12–72). A higher score in perceived physical ability scales is representative of a greater perceived physical ability, whereas greater scores on the physical self-presentation confidence are suggestive of an individual’s confidence in presenting their physical skills [21].

#### 2.3.4. Bench Press 1RM Specific Survey

Participants were asked to complete a survey consisting of 10 questions specifically created for this study for the purpose of analyzing how one’s self-efficacy on the BP may predict 1RM performance. Consistent with the survey by Ryckman et al. [21], each question is rated on a 6-point Likert scale. The survey was taken twice: (1) after initial agreement to participate in the study, and (2) the day of 1RM testing. If the test–retest intraclass correlation (ICC) between responses of the 1st and 2nd survey for all participants was 0.80 or greater, then the scores were to be used for analysis. However, the ICCs recorded were 0.55, and thus, participants’ PSE values rather than BP self-efficacy was used for analysis. It is speculated that because the second survey was performed immediately prior to BP 1RM testing, performance anxiety may have influenced participants’ self-efficacy and thus caused the discrepancy.

#### 2.3.5. Bench Press 1RM Protocol

The 1RM testing began with participants performing a 5 min general warm-up on a cycle ergometer at a pace equated to each participant’s perceived rating of “moderate intensity”. After the general warm-up, the lead investigator asked participants to predict their 1RM. Participants were instructed to lie on the bench (REP Fitness, Denver, CO, USA) with their head, back, shoulders, and buttocks in contact with the BP surface and the soles of both feet in contact with the ground. They were required to grip the bar (REP Fitness, Denver, CO, USA) with their hands with a “thumbs around” grip using a self-selected grip width. Participants had the option to un-rack the barbell themselves or receive assistance. After un-racking the barbell, participants were required to display control of the barbell with their elbows fully extended. The lead investigator then gave the participants a verbal “start” command where participants began to perform the eccentric portion of the lift by lowering the barbell to their chest or upper abdominal area. After the barbell made brief contact with the body (bouncing was not permitted), participants began the concentric portion by pressing the barbell up and fully extending their arms and holding the position for a brief “lock out”, after which the lead investigator gave the participants a verbal “rack” command. The 1RM assessment was consistent with guidelines established by the National Strength and Conditioning Association (NSCA) [22].

After the general warm-up, participants performed a specific warm-up consisting of 5–10 and 3–5 repetitions using 50% and 80% of estimated 1RM loads, respectively, with ~3 min recovery between each set. Participants then began to perform 1RM trials with ~5 min rest between attempts. The lead investigator used verbal feedback from the participant based on their rating of perceived exertion (RPE) [23] in conjunction with observation of barbell velocity during the lift to confirm whether another attempt should be made with additional load. A participant’s 1RM was established under one of three conditions: (1) a recording of an RPE of 10 by the participant or the lead investigator indicated that any increase in load would not be completed successfully, (2) an RPE of 9 or 9.5 followed by the participant failing the subsequent attempt of a load increase of <0.5 kg, or (3) a recorded RPE of <9 where the participant failed on the following attempt with a load increase of <0.5 kg. The lead investigator, a NSCA-certified strength and conditioning specialist (CSCS), was present to monitor all 1RM testing procedures.

#### 2.3.6. Dynamometry

A minimum of 48 h after initial 1RM testing, participants were asked to return to the lab to perform isometric dynamometry testing (Biodex Isokinetic Dynamometry System 4 Pro, Shirley, NY, USA). As previously mentioned, shoulder abduction and elbow flexion angles that were predicted for each participant’s sticking point were used for the dynamometry assessment. We obtained bilateral measurements of elbow extension and horizontal shoulder flexion (adduction) bilaterally (see image in Appendix A). For isometric horizontal shoulder flexion testing, participants were positioned lying supine with their torso and legs strapped securely to the chair using a safety belt to prevent trunk and lower body motion. The axis of rotation of the dynamometer arm was positioned superolateral to the glenohumeral joint such that the recorded net joint moment was in the direction of pure shoulder flexion. This positioning necessitated that the dynamometer arm be rotated based on the participants’ predicted shoulder extension and abduction angles. For elbow extension, participants were positioned seated in the dynamometer with their torso strapped securely to the chair with safety belts to prevent trunk motion. The axis of the dynamometer arm was oriented vertically with the chair adjusted so that each participant’s arm was at shoulder height when placed on the dynamometer arm. The length of the dynamometer arm was adjusted so that the rotation axis of the elbow was directly in line with the rotation axis of the dynamometer arm. Each trial lasted for 5 s, followed by a 30 s passive rest interval for 4 trials in each position. Participants were verbally encouraged to contract as hard as possible and were allowed to view the screen for biofeedback, which has been shown to increase performance of quadriceps and hamstring force production when compared to control [24]. The greatest peak net joint moments from each of the 4 trials for each position were used for analysis.

### 2.4. Statistical Analyses

All analyses were performed in R (version 4.2.1). A multiple linear regression was used to examine the contributions from the “primary” anthropometric, psychometric, and isometric strength variables to predict 1RM loads, as per our preregistration. The primary variables were selected based on previous studies [6,8,10,12] by combining (summing) variables contained within the categories of interest (anthropometric, biomechanical, and psychological). By combining highly correlated, conceptually similar variables into a single variable, the analysis maintained the effects of the multiple regression while allowing for the quantification of the relative contributions of each entire category of interest, since we were interested in drawing inferences concerning categories rather than individual variables. We qualitatively inspected the residuals to ensure model assumptions were met.

We performed two types of exploratory analyses. The first consisted of estimating the Pearson’s *r* correlation coefficients between all of the variables collected and BP 1RM. Confidence intervals were calculated using the bias-corrected and accelerated bootstrap with 9999 replicates. The second exploratory analysis used a combination of principal components analysis (PCA) and multiple regression, also known as principal components regression. PCA reduced our data’s dimensionality by combining variables that share variance—this was especially important in this study since we know a priori that many of our variables were related. To this end, we focused on upper body-related variables that we thought were more likely to relate to BP strength, as opposed to whole-body variables (e.g., FFM and skeletal muscle mass). In addition, we removed variables that were mathematically coupled to other variables in the dataset (e.g., ratios of limb lengths) from our data matrix **X**. Next, we column-wise *z*-scored **X** and performed PCA using singular value decomposition. To determine the number of dimensions to include, we performed parallel analysis, wherein we shuffled each column independently to destroy the correlation structure, re-ran PCA, and saved the eigenvalues. We kept the real principal components (PC) whose eigenvalues exceeded those from the parallel analysis. The final PCs were then used in multiple regression, along with sex as a covariate. Finally, we projected the regression coefficients back into variable space to improve interpretability.

## 3. Results

The average BP 1RM was 91 ± 19 kg. Multiple linear regression captured much of the variance in BP strength (F(3,14) = 5.34, *p* = 0.01; R^2^ = 0.53; adjusted R^2^ = 0.43) (Table 2, Figure 1). Total peak torque had the greatest standardized effect, followed by lean body mass and self-efficacy.

Bivariate Pearson correlations can be seen in Figure 2. Anthropometric and biomechanical measures tended to be strong and consistent predictors of BP strength while the psychological variables were largely centered around zero. Additionally, force plate data revealed a correlation coefficient of 0.476 (95% CI = −0.006–0.78), suggesting a moderate association between leg drive and BP strength.

PCA revealed one component above the noise floor (Figure 3A), in which all upper body anthropometric and biomechanical variables had positive loadings and PSE had a negative loading (Figure 3B). This single component accounted for 49% of the variance in BP strength (Figure 3C); the resulting weights can be seen in Figure 3D. The variance captured by the multivariable model and principal components regression was similar—their residuals (*r* = 0.76) and predictions (*r* = 0.77) were strongly correlated.

## 4. Discussion

The purpose of this study was to investigate the interplay between biomechanical, anthropometric, and psychological variables and their ability to predict BP strength among a heterogenous resistance-trained population. Multiple regression analysis revealed that biomechanical factors (total peak torque) best predicted BP performance, followed by anthropometric variables (FFM) and self-efficacy (Table 2, Figure 1). In agreement with previous work [10], bivariate correlations showed strong relationships between anthropometric variables and performance, and likewise for biomechanical variables. However, self-efficacy did not demonstrate an appreciable association with BP performance when considered in isolation (Figure 2). Multiple regression of lean body mass, total peak torque, and physical self-efficacy accounted for 43% of the variance in 1RM BP. Our principal components regression, containing primarily upper body anthropometric and biomechanical variables, accounted for a similar amount of variance using just a single component. The findings from multiple regression and bivariate associations showed that FFM was a moderate predictor of BP strength.

Our findings are slightly at odds with previous work that investigated BP performance [6,8,11]. While FFM is a strong proxy of muscle size, it consists of skeletal muscle mass, bone mass, and other organ tissues [6]. Perhaps more important, FFM is not specific to the upper body musculature relied upon in the BP—it is unlikely that leg FFM and the viscera are important contributors. This notion is supported by Figure 2, which shows that leg mass has a weaker correlation with BP strength than arm mass and trunk mass (*r* = 0.48 vs. ~0.68). The importance of upper body FFM for success in BP performance is further supported by the strong correlations between BP 1RM and relaxed and flexed upper arm circumferences (Figure 2), which is supported by prior research [13].

Regarding isometric strength, total peak torque (isometric horizontal shoulder flexion + isometric elbow extension) had the greatest standardized effect on maximal BP strength. Although the strong association observed suggests that it would be prudent for individuals to include single-joint training of the horizontal shoulder flexors and elbow extensors to help improve BP strength, the complexity of the BP does not allow insight as to which of the prime movers are the limiting factors to performance, as the maximum isolated joint strength may not necessarily reflect the demands during the BP. Since isometric strength testing is performed in a controlled environment where the joint is isolated, co-contraction and coordination requirements are limited. This idea is reinforced by comparing our isometric net joint moments to those estimated in the BP [20]. The maximum isometric horizontal shoulder flexion and elbow extension observed were greater than the highest joint horizontal shoulder (197 vs. 184 N·m) and elbow (167 vs. 88.6 N·m) moments from Larsen et al. [20], whose participants were heavier (87.8 vs. 80.4 kg) and used greater loads at various grip widths (110 kg wide, 109 kg medium, 104 kg narrow vs. 91 kg in our study). Thus, much like findings observed by Vigotsky et al. [10] in the squat, BP performance may amount to less than the sum of individual parts, at least in part due to the BP involving dynamic effort with more mechanical degrees of freedom. In contrast to our isometric net joint moments, the net joint moments reported by Larsen et al. [20] describe the minimum muscular effort—or sum of all the muscles—where co-contraction was likely a factor. For example, the long head of the triceps brachii can extend the shoulder [25], which is counterproductive to generating horizontal shoulder flexion moments during the BP. This is analogous to the hamstrings in the squat, which can generate hip extension moments, but at the cost of producing knee flexion moments, which the quadriceps must then overcome [26].

Corroborating the previous work by Vigotsky et al. [10] in the squat, PSE had little association with BP strength. Indeed, self-efficacy is a complex phenomenon and is predicated on previous experience. As one gains more exposure to strength training and develops technical proficiency, it is expected that their confidence in their ability to BP should increase. Thus, PSE may be better able to explain the variance in strength within a single subject or across a homogeneous sample; however, we employed a heterogeneous sample. While self-efficacy may increase an individual’s ability to involve their musculature, it does not necessarily increase their potential to generate a joint moment [10].

This study has several limitations that should be considered when drawing inferences. For one, although we employed a heterogeneous population, the sample was largely men, and thus caution needs to be warranted when extrapolating the results to women [27]. Moreover, this study investigated resistance-trained college-aged students who had a minimum of 1 year of training experience and had been bench pressing at least once per week for the past 6 months; thus, the results reflect the performance of the general recreational lifting population and cannot necessarily be extrapolated to more advanced trainees such as powerlifters or elite athletes. In addition, individual sticking point predictions were performed using a model based on the data of nine resistance-trained males in conjunction with participants’ barbell kinematics and upper arm lengths. This created an assumption of a homogenous sticking point and joint angles, which is likely less accurate than using 3D motion capture. However, this approach was necessitated due to a limited number of motion capture cameras, which created significant gaps and spurious marker locations that impaired the ability to accurately assess participants’ joint angles. Regarding self-efficacy, although the PSE survey has been validated in previous literature, it is not specific to the BP and thus fails to account for task specificity. Although an additional survey was created to resolve this limitation, the ICCs from the survey were too low to warrant inclusion in the analysis. Since the second BP survey was conducted directly before 1RM testing, pre-testing anxiety may have played a role in the test–retest discrepancies. Finally, participants were not allowed to wear wrist wraps or a belt during their 1RM testing, which may have influenced their self-efficacy.

## 5. Practical Applications

Our data indicate that BP strength among a heterogenous resistance-trained population is primarily predicated on biomechanical and anthropometric variables, with psychological variables playing a minor role. From a practical standpoint, these findings shed light on factors that may influence BP strength, which may help practitioners improve training regimens. For example, training regimens aiming to improve BP performance should focus most on training to increase the lifter’s ability to horizontally adduct the shoulder with the pectoral muscles and extend the elbow with the triceps. Additionally, training to build and accrue FFM should be a high priority, while focusing on the development of self-efficacy factors should be of minor concern. Lastly, the moderate association between leg drive and BP suggests that developing technical proficiency to increase force from the feet during the BP may improve performance.

Scientists can use these exploratory results to further understand how these cross-sectional measures may causally affect and or longitudinally change with BP strength. For instance, perhaps a set-equated study comparing a bench press alone versus additional isolated exercises for horizontal shoulder adduction (i.e., pectoralis fly) and elbow extension (i.e., triceps press down) versus bench press alone would reveal the benefit of incorporating specific isolation exercises into a training routine or highlight the necessity to focus on specificity when aiming to increase strength in a specific exercise such as the BP.

## Figures and Tables

**Figure 1 sports-10-00199-f001:**
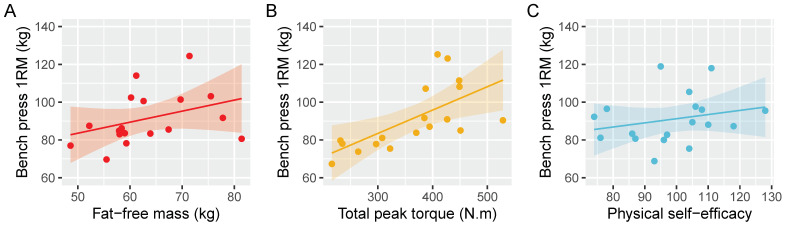
Linear regression results depicting the adjusted marginal relationships between bench press 1RM and our three primary independent variables. (**A**) After adjusting for total peak torque and physical self-efficacy, fat-free mass has a positive but uncertain relationship with bench press strength. (**B**) In contrast, total peak torque has a consistent, positive relationship with bench press strength. (**C**) Like fat-free mass, physical self-efficacy was estimated to have a positive but uncertain slope.

**Figure 2 sports-10-00199-f002:**
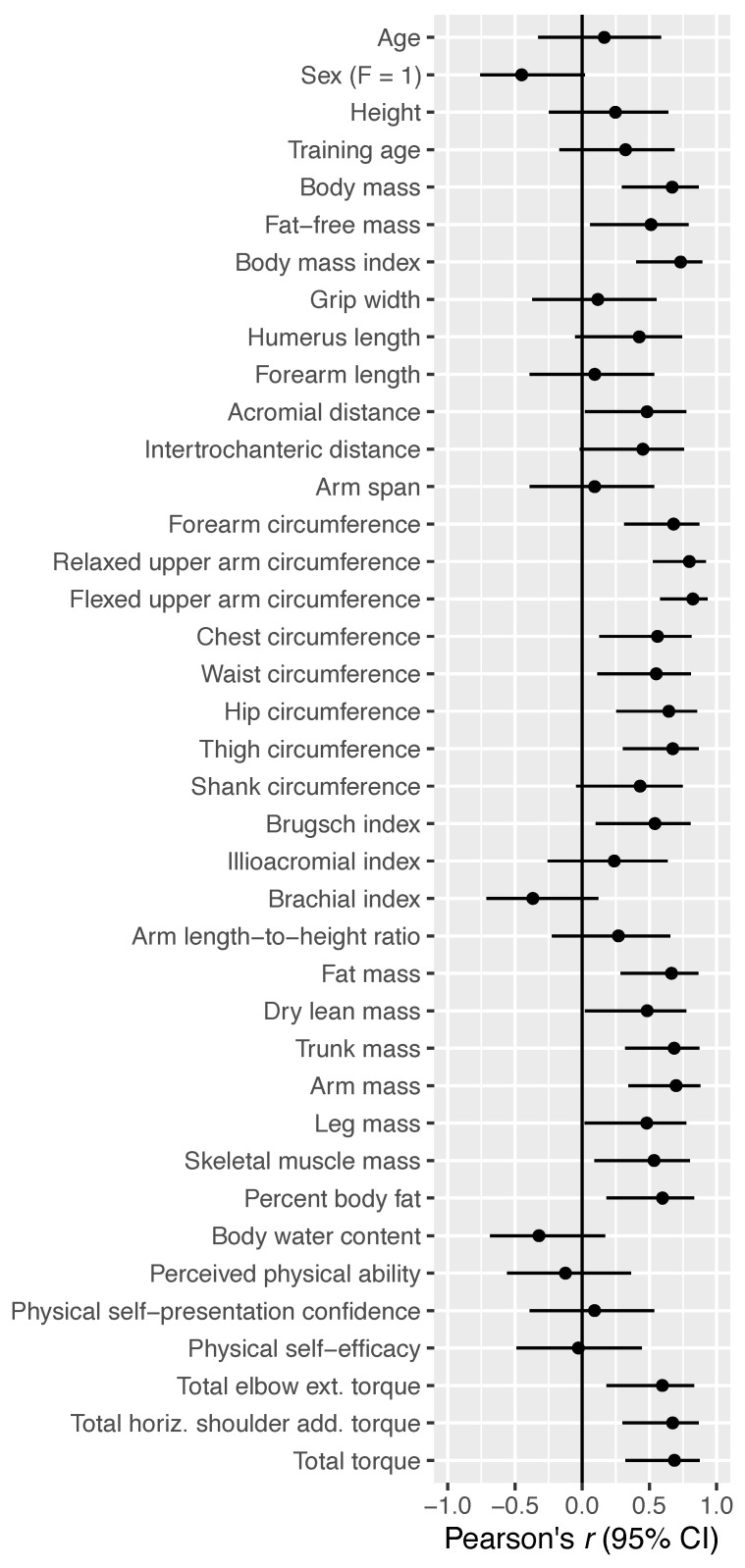
Correlations between determinants of bench press strength with bench press 1RM.

**Figure 3 sports-10-00199-f003:**
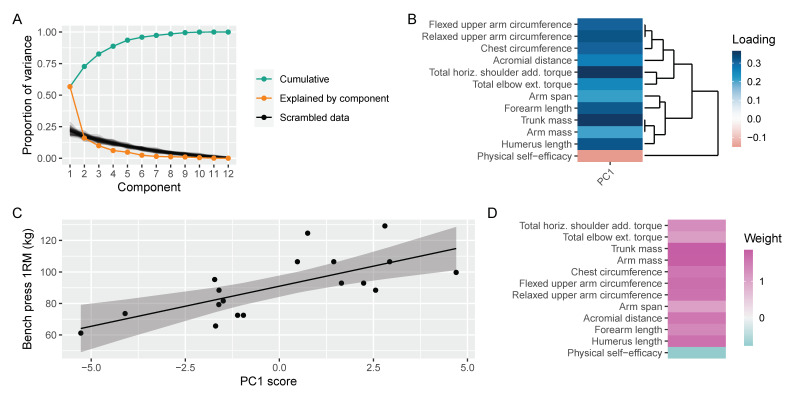
Principal components regression captured a similar amount of variance as our multivariable model using only a single component. (**A**) Parallel analysis revealed that only a single component was above the floor. (**B**) Anthropometric and biomechanical variables had positive weights, while PSE had negative weight. (**C**) The first principal component accounted for 48% of the variance in BP strength. (**D**) We projected the regression weight back onto the variables.

**Table 1 sports-10-00199-t001:** Participant demographics.

Sex	*n*	Age (y)	Body Mass (kg)	Height (m)
Female	2	28 ± 4	62 ± 2	1.62 ± 0.02
Male	16	22 ± 5	83 ± 14	1.74 ± 0.06
Combined	18	23 ± 5	80 ± 15	1.73 ± 0.07

Data are mean ± SD.

**Table 2 sports-10-00199-t002:** Results of multiple linear regression on determinants of barbell BP strength.

	Estimate ± SE	95% CI	*t* Value	*p*-Value	Standardized Coefficients
Intercept (kg)	91.0 ± 3.4	83.7–98.2	26.94	<0.001	
FFM (kg/kg)	0.58 ± 0.47	−0.42–1.59	1.245	0.2334	0.27
Total peak torque (kg/(N·m))	0.12 ± 0.04	0.03–0.22	2.749	0.0157	0.59
Self-efficacy (kg/PSE)	0.22 ± 0.25	−0.32–0.76	0.879	0.3942	0.17

Abbreviations: kg = kilograms; FFM = fat free mass; N·m = newton-meters; PSE = physical self-efficacy.

## Data Availability

Data can be obtained at: https://osf.io/d4qa2 (accessed on 22 February 2022).

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
