# Peer review of "Biomechanical, Anthropometric and Psychological Determinants of Barbell Bench Press Strength"

_sports, 2022, doi:10.3390/sports10120199_

Round 1

Reviewer 1 Report

There are some typo:

- At line 51: "(see: 10)";

- In Figure 3 there is a double reference to panel "C" and no reference to panel "D"

- In Table  2: why the intercept has the unit of measure?

- And why FFM is: "Kg/Kg" and not just "Kg"?

That said, the article is well constructed, but the sample is modest. As mentioned in the discussion the sample is not gender homogeneous and so the results are hardly relevant to women.

I think that some drawings or shoots of the kinematic measures and Dynometry testing could help.

Author Response

There are some typo:

- At line 51: "(see: 10)";

- In Figure 3 there is a double reference to panel "C" and no reference to panel "D"

- In Table  2: why the intercept has the unit of measure?

- And why FFM is: "Kg/Kg" and not just "Kg"?

AUTHOR RESPONSE: Thank you for your positive feedback and for pointing out the typos. We have revised the manuscript to correct all applicable instances that you mention. Note that the intercept refers to the expected y if all of the xs are zero. Since y has units kg, the intercept has units kg. Also, the kg/kg for FFM represents its relationship with BP.

That said, the article is well constructed, but the sample is modest. As mentioned in the discussion the sample is not gender homogeneous and so the results are hardly relevant to women.

AUTHOR RESPONSE: Yes, we agree and have listed this as a limitation.

I think that some drawings or shoots of the kinematic measures and Dynometry testing could help.

AUTHOR RESPONSE: As per your request, we have uploaded an image of the dynamometry positions to supplemental files.

Reviewer 2 Report

This research once again demonstrates the primary role of anthropometrics and biomechanics in bench press performance. It reports a new, multi-integrated approach to the kinanthropometric analysis of bench press. This paper is well written, and the research protocol is solid and complete. Comprehensively, hypothesis are properly verified, statistical analysis are generally complete -although classic multiple regression analysis would have been fine-. Results are consistent with the discussion. The set of anthropometric measurement is complete and well-selected.

There are only a few remarks/observation I can point out; although, these observations are more likely to be taken in considerations for authors' next study:

1) the sample number is low. Bias may have been introduced due to the scarcity of the sample and to the fact that males and females were considered in the same group. Nevertheless, authors themselves point out how this research is an exploratory study; in this perspective, the point is overall acceptable.

2) the sample doesn't consist in elite athletes, nor competitive amateur athletes, through which results may have been more standardized. Again, this point is mentioned by authors as a limit of their study.

3) for a future perspective, we suggest to adopt plicometry as an addictional methodology for body composition assessment, since BIA  can be often be influenced by external factors.

Author Response

REVIEWER 2

This research once again demonstrates the primary role of anthropometrics and biomechanics in bench press performance. It reports a new, multi-integrated approach to the kinanthropometric analysis of bench press. This paper is well written, and the research protocol is solid and complete. Comprehensively, hypothesis are properly verified, statistical analysis are generally complete -although classic multiple regression analysis would have been fine-. Results are consistent with the discussion. The set of anthropometric measurement is complete and well-selected.

There are only a few remarks/observation I can point out; although, these observations are more likely to be taken in considerations for authors' next study:

1) the sample number is low. Bias may have been introduced due to the scarcity of the sample and to the fact that males and females were considered in the same group. Nevertheless, authors themselves point out how this research is an exploratory study; in this perspective, the point is overall acceptable.

2) the sample doesn't consist in elite athletes, nor competitive amateur athletes, through which results may have been more standardized. Again, this point is mentioned by authors as a limit of their study.

3) for a future perspective, we suggest to adopt plicometry as an addictional methodology for body composition assessment, since BIA  can be often be influenced by external factors.

AUTHOR RESPONSE: Thank you for the time and effort reviewing our paper. Yes, we had hoped to get a larger sample of females, but it ultimately was not feasible based on the response from the available population. As you noted, we mentioned this as a limitation. Your point is noted about the addition of plicometry; we will employ in future research when applicable.

Reviewer 3 Report

This study is based on dynamic/kinematic analyses and psychological evaluations to investigate the biomechanical, anthropometric and psychological parameters of maximal bench press strength in a heterogeneous resistance-trained sample. One of the strengths of this study is the statistical analysis. It is a study of good quality which requires some corrections.

1)    Page 1, line 22 : the variance of BP : Abbreviations should be avoided in the abstract or preceded by a full sentence (barbell bench press).

2)    Page 3, line 122 : it should be noted that the anthropometric measurements were taken by the same experimenter.

3)     A statistical technique (Principal Component Analysis) was used to to reduce the number of data collected and determine the structure of the relationships between these collected data and also proceeded to detect and eliminate "outliers" by Z-scored the initially recruited subjects.

As a first step in PCA, the relationship between the variables should be examined by Bartlett's test of sphericity.

- Have you applied the "KMO" test: Kaiser-Meyer-Olkin Index?

- Have you checked the results and the factors to be retained by the parallel analysis with the cone diagram?

4)    Can you include in the Materials and Methods paragraph one or two pictures relating to the experimental protocol (BP)?

5)    Page 10, line 413 : leave a space before and after the semicolon punctuation. To be adopted for all references.

Author Response

REVIEWER 3

This study is based on dynamic/kinematic analyses and psychological evaluations to investigate the biomechanical, anthropometric and psychological parameters of maximal bench press strength in a heterogeneous resistance-trained sample. One of the strengths of this study is the statistical analysis. It is a study of good quality which requires some corrections.

AUTHOR RESPONSE: Thank you for the detailed feedback. We have addressed your comments on a point-by-point basis below, and made revisions to the manuscript (highlighted in red) where applicable.

  • Page 1, line 22 : the variance of BP : Abbreviations should be avoided in the abstract or preceded by a full sentence (barbell bench press).

AUTHOR RESPONSE: Good catch. We have revised accordingly

  • Page 3, line 122 : it should be noted that the anthropometric measurements were taken by the same experimenter.

AUTHOR RESPONSE: Yes, good point. We have revised to note that the measurements were obtained by the same investigator.

3)     A statistical technique (Principal Component Analysis) was used to to reduce the number of data collected and determine the structure of the relationships between these collected data and also proceeded to detect and eliminate "outliers" by Z-scored the initially recruited subjects.

As a first step in PCA, the relationship between the variables should be examined by Bartlett's test of sphericity.

- Have you applied the "KMO" test: Kaiser-Meyer-Olkin Index?

- Have you checked the results and the factors to be retained by the parallel analysis with the cone diagram?

AUTHOR RESPONSE: These are implicit in the parallel analysis. If the correlation matrix was consistent with the identity matrix (i.e., this relates to what KMO and Bartlett evaluate), the parallel analysis would reveal that even the first component is consistent with the noise model. Nevertheless, KMO = 0.69 and Bartlett’s chi-squared(df=66) = 256, P = 3e-24. We’d prefer to leave this out of the manuscript since these are typically more common for factor analysis (not PCA) and are also sensitive to distributional assumptions.

Could you elaborate on what you mean by a “cone diagram”?  Are you referring to a biplot?

4)    Can you include in the Materials and Methods paragraph one or two pictures relating to the experimental protocol (BP)?

AUTHOR RESPONSE: Unfortunately data collection has already concluded and we did not take photos, so this is not possible.

5)    Page 10, line 413 : leave a space before and after the semicolon punctuation. To be adopted for all references.

AUTHOR RESPONSE: The reference formatting is carried out by the journal so this will be corrected upon publication.  
